# Research on Green View Index of Urban Roads Based on Street View Image Recognition: A Case Study of Changsha Downtown Areas

Yixing Chen [1,2,*], Qilin Zhang [1], Zhang Deng [1], Xinran Fan [3], Zimu Xu [3], Xudong Kang [3], Kailing Pan [3] and Zihao Guo [3,4]

1    College of Civil Engineering, Hunan University, Changsha 410082, China
2    Key Laboratory of Building Safety and Energy Efficiency of the Ministry of Education, Hunan University, Changsha 410082, China
3    School of Architecture and Planning, Hunan University, Changsha 410082, China
4    School of Architecture and Planning, The University of Auckland, Private Bag 92019, Auckland 1142, New Zealand
*     Correspondence: yixingchen@hnu.edu.cn

**Abstract:** In this paper, we took the urban roads in the Changsha downtown areas as an example to identify the green view index (GVI) of urban roads based on street view images (SVIs). First, the road network information was obtained through OpenStreetMap, and the coordinate information of sampling points was processed using ArcGIS. Secondly, the SVIs were downloaded from Baidu Map according to the latitude and longitude coordinates of the sampling points. Moreover, semantic segmentation neural network software was used to semantically segment the SVIs for recognizing the objects in each part of the images. Finally, the objects related to green vegetation were statistically analyzed to obtain the GVI of the sampling points. The GVI was mapped to the map in ArcGIS software for data visualization and analysis. The results showed the average GVI of the study area was 12.56%. An amount of 27% have very poor green perception, 40% have poor green perception, 19% have general green perception, 10% have strong green perception, and 4% have very strong green perception. In the administrative districts, the highest GVI is Yuhua District with 14.15%, while the lowest is Kaifu District with 8.75%. The average GVI of the new urban area is higher than that of the old urban area, as the old urban area has higher building density and a lower greenery level. This paper systematically evaluated the levels of GVI and greening status of urban streets within the Changsha downtown areas through SVIs data analysis, and provided guidance and suggestions for the greening development of Changsha City.

**Keywords:** street view image; green view index; image identification; semantic segmentation

## 1. Introduction

With the gradually accelerating process of urbanization, more than 56% of the world-wide population lives in urban areas after 2020 [1]. In the past decade, social attention has shifted from quantity demand to quality demand, which makes urban greening research play an important role in sustainable urban development [2]. People are paying more and more attention to the livability of cities [3] as well as the quality of urban street space.

To facilitate sustainable urban development in a better way, a variety of indicators need to be applied to the evaluation system. Among them, urban greening is an essential indicator in the evaluation of urban livability [4]. Urban roadway greenery (trees, shrubs, lawns, and other forms of vegetation lining the street) is a critical design element in the urban green landscape and has effects on pedestrians in a variety of aspects, including human visual comfort, wind comfort, thermal comfort and their stress levels [5–7]. In the follow-up study, we mainly target the pedestrians visual comfort aspects.

At present, there are various indicators used in some of the case studies to evaluate urban greenery, including the Green Coverage Ratio (GCR), Building Visual Greenness Index (BVGI), Baidu Green View Index (BGVI), Green View Index–Urban Transportation (GVI-UT), and Urban Neighborhood Green Index (UNGI) [8–12]. Current research on the association of urban street quality with the GVI is mainly concerned with the following aspects: definition and significance of the GVI, exploring urban street quality based on urban street view images (SVIs) and machine learning algorithms, and investigating the spatial distribution characteristics of the urban green view index based on street view recognition.

The GVI is a commonly used green amount indicator, and is distinct from the traditional green amount evaluation, which has been an official regular indicator of the green landscape evaluation system in Japan [13,14]. Xiao et al. [15] proposed that the application of the GVI is conducive to promoting urban greenery construction in China by outlining the research findings of Japanese scholars on the GVI in the past 30 years. The most critical point is that the GVI is the key cognitive factor in the research of street green space, and can evaluate urban greenery quantitatively, so as to compensate for the shortcomings of traditional greening indicators [16,17].

Along with the development of computer technology and big data analysis, image processing techniques could be applied to urban-scale studies, such as dating the construction year of building groups and analyzing street vitality [18,19].

SVIs and machine learning are gradually used in the research of urban street quality. Researchers could acquire SVIs for research using urban street view services from some map systems, such as Google, Baidu, and Tencent Map [20]. Cui et al. [21] used SIFT methodology to distinguish SVIs of the fall and winter in Harbin; the study mainly focused on the analysis of the superposition of different urban zoning districts with GVI. Li et al. [22] conducted a study on the relationship between the spatial distribution of street greenery and several socio-economic variables in the residential area of Hartford, Connecticut, USA, and the results indicated that people with different social conditions live in streets with different levels of greenery. Ye et al. [23] captured a large amount of SVIs data using Baidu Map API, applied an image separation algorithm based on machine learning to calculate the proportion of greenery in the street view, and used a spatial network to analyze street accessibility, which could provide detailed and scientific guidance for urban greening planning and design. Machine learning can also be applied to the classification of vegetation in SVIs with high accuracy [24].

Over recent years, large-scale automated SVIs processing technology has been gradually utilized [25]. The street view data analysis based on deep learning is a useful method for automatically evaluating streetscapes [26]. Zhang and Hu [27] analyzed the GVI on a large scale with deep learning, and demonstrated that it is a feasible method to calculate the green view index best path. Meanwhile, the combination of deep learning and semantic segmentation improves the efficiency of detection and classification of targets [28]. Shao et al. [29] picked the streets within the fifth ring road in Beijing and the outer ring road in Shanghai to conduct the study, extracted the streetscape data through the Baidu Map platform, classified the streetscape element composition using the image semantic segmentation technology, and then assessed through the network to determine which type of streetscape elements are positive or negative on the streetscape comfort.

The previous urban-scale greening studies, such as urban trees classification and urban greenspace change detection, typically relied on satellite and aerial imaging techniques [30,31]. However, the SVI identification technology is gradually maturing and has been applied to many of the present studies, such as screening for specific types of buildings and visual place recognition [32,33]. Xu et al. [34] calculated the green pixels to derive the street GVI using OpenCV, a Python image processing library, for 98 roads in the old town of Zhengzhou city, and then visualized spatial distribution with analysis and evaluation. Wang et al. [35] used the SegNet method to recognize the greenery in numerous SVIs to obtain the data of panoramic green view index (PGVI), and analyzed the contributing factors of PGVI to provide new thoughts for the optimization research of

urban greenery in the future. Thus, the SVI identification has led to the improved accuracy and efficiency on the GVI of urban roads research.

To obtain the data of the key contributing factors on urban street greenery, such as tree cover, the computer vision algorithms can be used to segment and quantify tree cover in a large amount of SVIs, which contributes to the quantitative research on the spatial distribution of urban greenery from the perspective of citizen experience and urban landscape observation [36]. Li et al. [37] utilized the trained deep convolutional neural network model based on street images to identify different urban characteristics; the study measured the spatial distribution of street greenery in Cambridge, Massachusetts, providing a fresh perspective for urban landscape studies worldwide. The full convolutional neural network (FCN) is the development basis for many existing semantic segmentation methods and its model can be coupled with urban scene elements for deep learning datasets [38]. On the basis of which the data in the urban landscape environment can be accurately identified at large scale and high granularity, therefore, by overlaying with the spatial data, the complex-built environment of the city can be accurately analyzed and finely repaired [39].

In summary, the accuracy and efficiency could be improved by recognizing the GVI of urban streets based on SVIs, which has received much attention of scholars all around the world. In this paper, we took the partial urban roads within the second ring in Changsha (as shown in Figure 1) as an example for our study, and conducted research on the GVI of urban roads based on SVI identification. Our research contents are as follows: (1) Obtaining urban road information. (2) Acquiring SVIs with specified coordinates. (3) Semantic segmentation of SVIs. (4) Calculating the GVI of coordinate points. (5) The GVI analysis in different urban roads and different areas. Finally, we objectively and systematically evaluated the level of GVI and the current status of greenery in the urban streets within the Second Ring of Changsha, as well as made reasonable suggestions for urban construction.

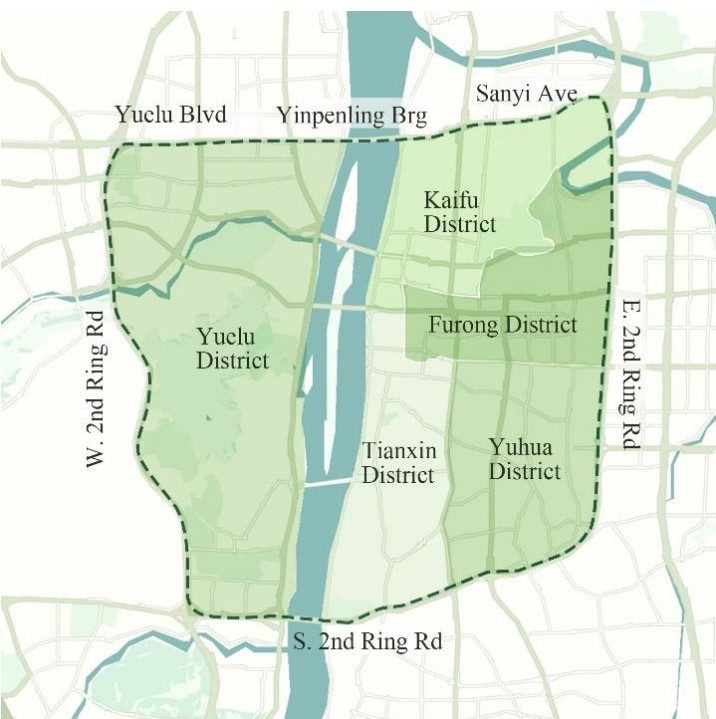

**Figure 1.** Map of the study range.

This study has mainly focused on the following objectives: (1) Use the GVI as an indicator to quantify the current urban roads greening in Changsha City. (2) Form a complete greening evaluation method, contributing to the development of the urban greening evaluation system. (3) Propose reasonable and feasible suggestions for the overall urban greening evaluation results.

## 2. Methods

Figure 2 shows our research methodology and technical route. Firstly, according to the significance and background of the study, the preliminary feasibility study was conducted based on the comprehension of relevant literature, and made clear the analysis range of the data required. Then, the Baidu Map SVIs data were captured by the tools, and the semantic segmentation software is used to obtain the GVI data of each sampling point. Finally, by applying ArcGIS to overlay the data of GVI and spatial geography, we can visualize and analyze the data to get an accurate diagnosis and resulting feedback, so as to give suggestions and strategies for GVI optimization.

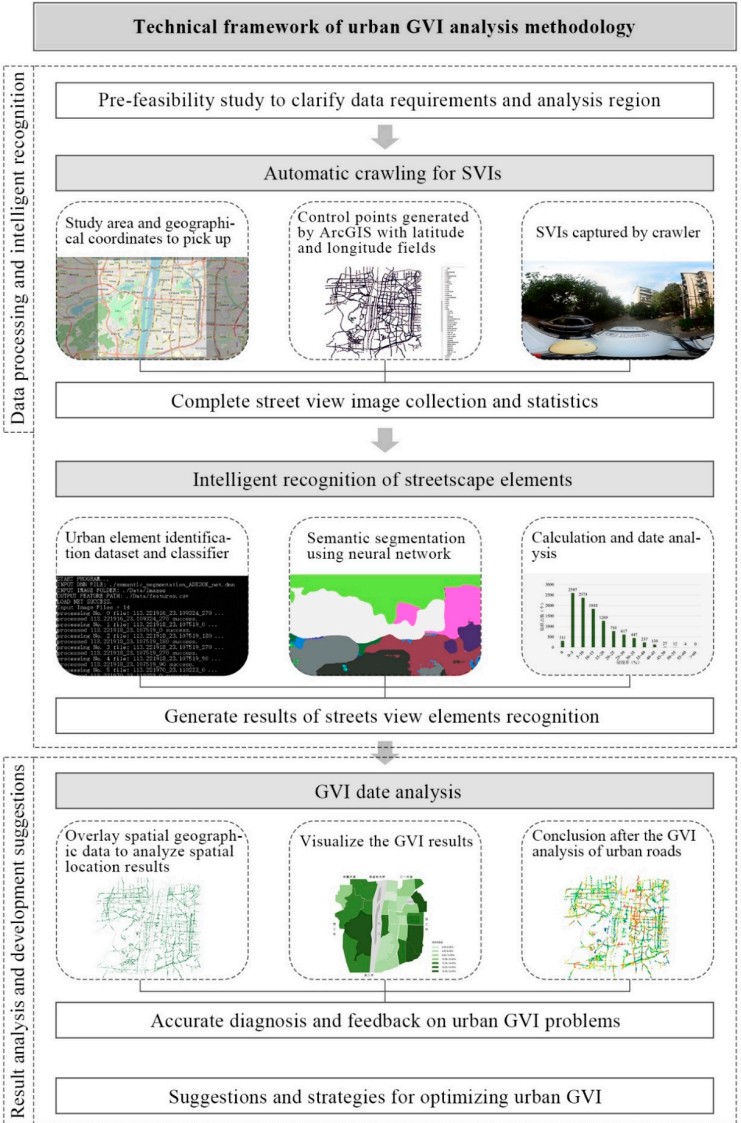

**Figure 2.** Technology roadmap.

### 2.1. To Obtain the Data of SVIs

The research adopted a method that could accurately acquire the SVIs data in batches. The data of the vector road network can be obtained through the OpenStreetMap website [40]. With the data above, ArcGIS is used to generate sampling points in the area and call the Baidu Street View Image Application Program interface to obtain SVIs data according to the latitude and longitude coordinates of the sampling points. The flow for obtaining data is shown in Figure 3.

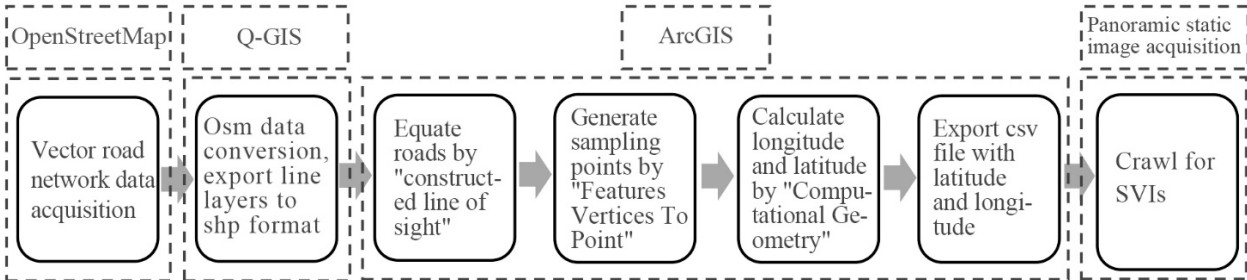

**Figure 3.** Steps to get SVIs data.

### 2.1.1. To Acquire Vector Road Network Data

The urban roads network data was downloaded from OpenStreetMap, an open-source website, and the result was shown in Figure 4. This study area is within the orange dotted line, including different administrative regions and planning zones (the core areas of Furong District, Tianxin District, Kaifu District, Yuhua District, and Yuelu District) in both new and old regions of Changsha City. Roads in each region are abundant in type and have clear boundaries, which facilitates comparative analysis.

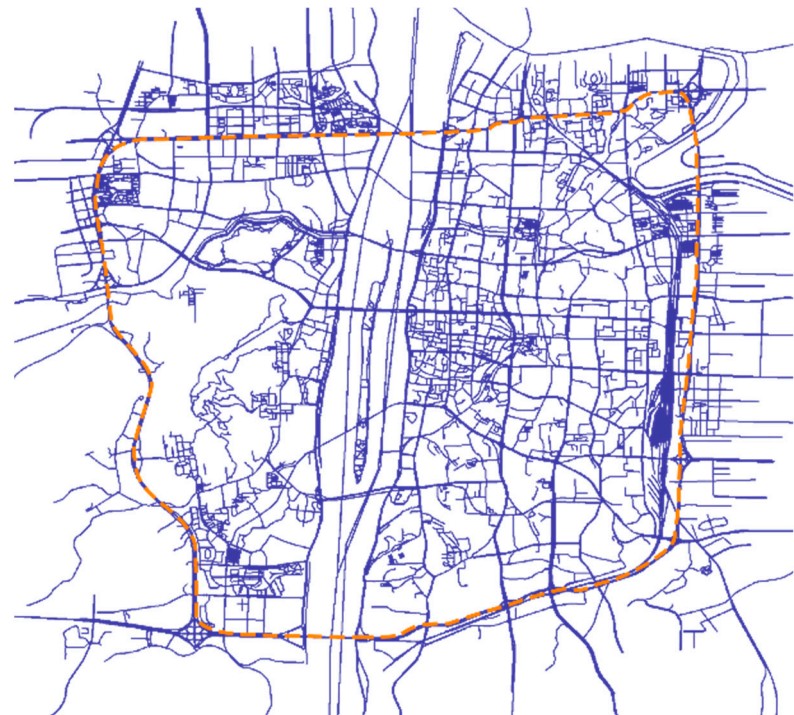

**Figure 4.** The study range of road network data.

### 2.1.2. Generate Control Points Randomly on the Road Vector Map of the Selected Area Using ArcGIS

In order to ensure the integrity of roads and facilitate the statistics and data analysis, the atypical roads, such as residential, path, footway, link, were excluded. Considering pedestrian's effective visual distance on a flat street and the amount of data statistics, the minimum distance between any two random sample points was set as 200 m by using the "Constructed Line of Sight" tool to equally divide the road length. Then, we used the tool "Features vertices To Point" to generate sampling points, and deleted the sample points that were too close to each other. Finally, 16,688 sample points were obtained for the study. Figure 5 shows an example of generated road sampling points using ArcGIS.

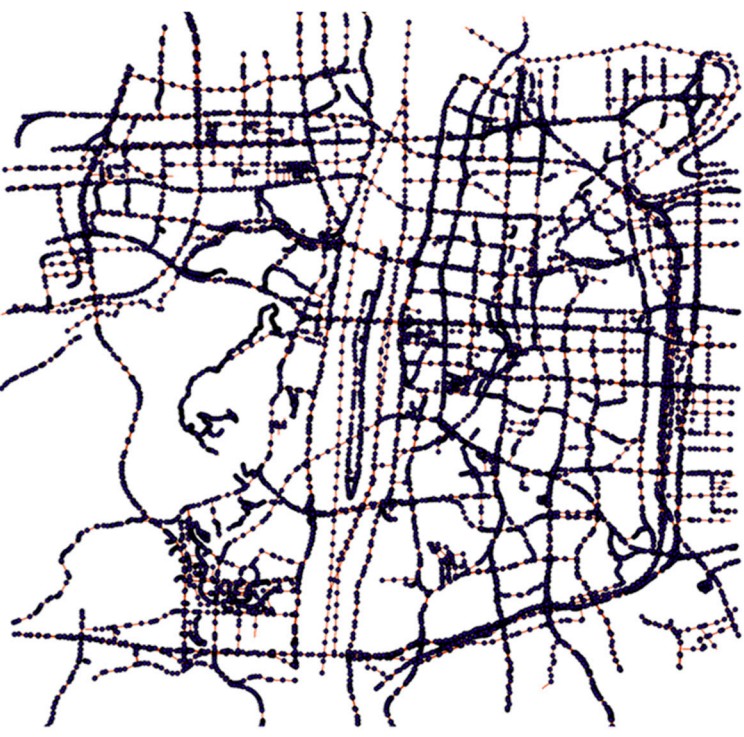

**Figure 5.** Road sampling points generated by ArcGIS.

### 2.1.3. Obtaining Latitude and Longitude Data of Road Sample Points

The data of latitude and longitude of each point were calculated by ArcGIS "Calculate Geometry" and saved as csv files to be used as input data for subsequent SVIs retrieval.

### 2.1.4. Obtaining SVIs Data

The Baidu Panorama Static Image Acquisition Tool can be used to call the Baidu Street View Image Application Program Interface (API) in order to obtain SVI data. The parameters of the viewing angle range are set to 360°, while the visibility range is a spherical environmental vision covering the horizontal and vertical directions of each sample point, so that the pedestrian perspective can be better simulated. In addition, a few sampling points not covered by Baidu Street View are deleted to ensure data accuracy. Finally, the data of 10,659 SVIs were downloaded and stored for subsequent green vegetation identification. The image pixel is 1024 × 512, with good image quality, which could show the green visual range more clearly and intuitively. Figure 6 shows the Baidu SVIs obtained from the above process, taken in August 2019. It is necessary to point out that the fish-eye type 360° images may differ slightly from human vision and thus impacts the accuracy of GVI calculation. However, the 360° images can capture the greenery information from all orientations and avoid the image overlap problem.

### 2.2. The GVI Analysis of SVIs

### 2.2.1. Extraction of Street View Elements

In this study, we extracted various elements in SVIs and obtained the data of the percentage of each element with the help of image semantic segmentation technology. Deep learning full convolutional network (FCN) based on ADE_20K dataset trained by Yao et al. [41] was used for SVI semantic segmentation in our research. Image semantic segmentation is "the ability to separate the foreground from the background in an image and identify each foreground target, assigning a category to each region (or pixel), which is equivalent to assigning a semantic label to each pixel" [42]. The image semantic segmentation software can classify the visual elements of SVIs into 150 categories, including building, wall, sky, tree, flora, ground, vehicle, sidewalk, and carriageway. The results generated

by image semantic segmentation can be displayed symbolically in ArcGIS, as shown in Figure 7, with each type of element distinguished by a different color. The boundaries of the different colors are clear and match the original image well.

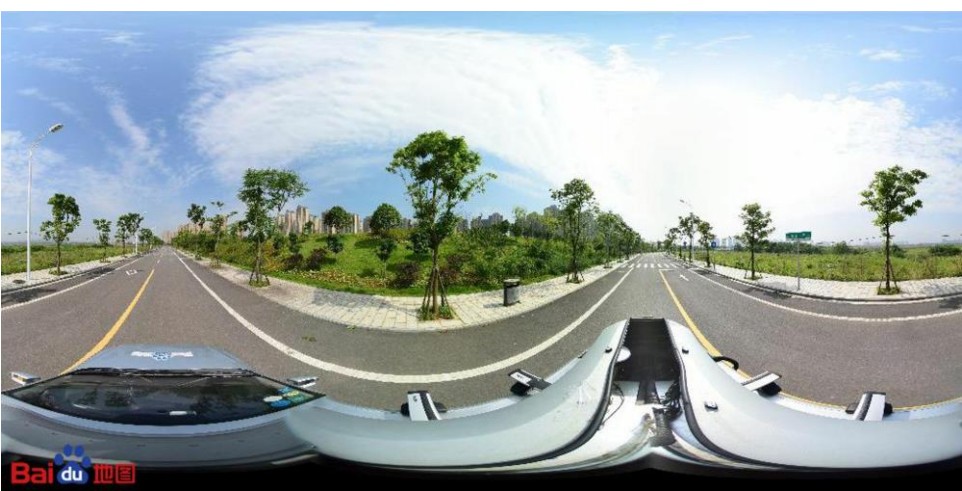

**Figure 6.** An example of Baidu street view image.

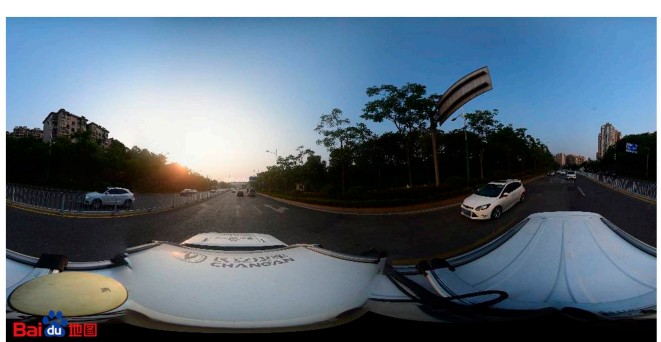

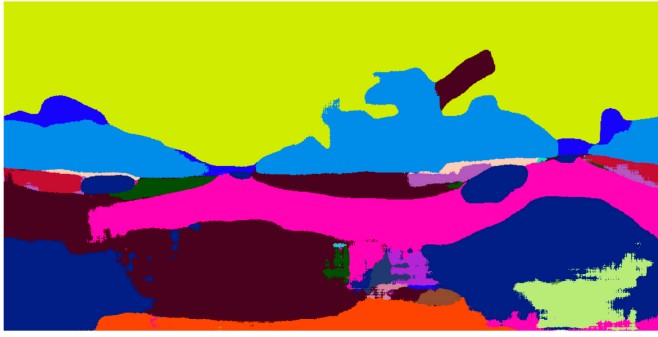

(**a**) The image of street view
(**b**) The image processed by semantic segmentation

**Figure 7.** Comparison of images generated by semantic segmentation and street view panorama.

Simultaneous statistics for the percentage of each element type in SVIs can be implemented with the image semantic segmentation software, and the results can be saved as CSV format files. In this paper, we selected three elements related to GVI: tree, grass, and vegetation, and then calculated the GVI of each image. The results are shown in Table 1.

**Table 1.** Analysis results for 3 types of elements (partial).

| Image Number | (Longitude, Latitude) | Element Name | | | GVI |
|---|---|---|---|---|---|
| | | Tree | Grass | Vegetation | (%) |
| 13,364 | (112.9433483, 28.2257947) | 6.71% | 0.39% | 0.27% | 7.37% |
| 13,365 | (113.0144491, 28.225649) | 6.81% | 0.08% | 1.91% | 8.80% |
| 13,366 | (113.0142805, 28.2257309) | 8.87% | 0.76% | 1.71% | 11.34% |
| 13,367 | (112.9442002, 28.2258079) | 17.63% | 0.11% | 2.11% | 19.86% |
| 13,368 | (112.9445804, 28.2258138) | 4.14% | 0.08% | 0.04% | 4.26% |
| . . . | . . . | . . . | . . . | . . . | . . . |

### 2.2.2. GVI Data Visualization

The aggregated GVI data was imported into ArcGIS and visualized in three levels: the whole Second Ring area in Changsha City, Hedong and Hexi regions, and each street within the district administrative unit, respectively. In the overall map, the numerical value of GVI was presented as a color gradient to be distinguished, while for each street in Hedong and Hexi regions, they were presented in the form of color blocks. As for the streets in the regional administrative unit, the GVI statistics for each road were aided by administrative planning maps, and only sample points of major streets within the Second Ring of each district were selected as the data to be exported. Then, the GVI value of each street was calculated by statistics and analysis. Finally, the numerical value was mapped to the map and the feedback was transformed into a graphical color block representation.

## 3. Result Analysis

### 3.1. Overall Numerical Characteristics of GVI for Some of the Roads within the Second Ring in Changsha City

In this study, a total of 10,659 sampling points of urban roads were collected from partial roads within the Second Ring in Changsha City, and the frequency histogram of GVI at the sampling points is shown in Figure 8. According to the classification standard used in recent relevant studies [43,44], the GVI can be divided into five levels: 0–5% (very poor green perception), 5–15% (poor green perception), 15–25% (general green perception), 25–35% (strong green perception), ≥35% (very strong green perception). Overall, the average GVI of partial roads within the Second Ring in Changsha is 12.56%, which is at a very poor level of green perception. The number of sample points with GVI in 0–5% and 5–10% are the largest, which are 2597 and 2374, respectively, accounting for 67% of the total, and greatly lower than the overall average. Of the streets in the study area, 27% have very poor green perception, 40% have poor green perception, 19% have general green perception, 10% have strong green perception, and 4% have very strong green perception.

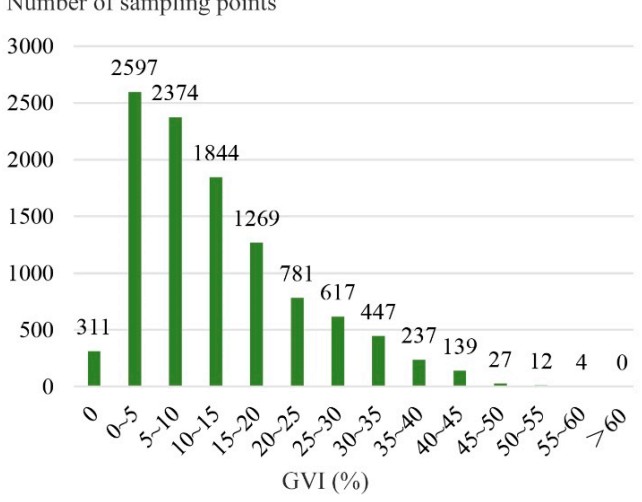

**Figure 8.** Distribution interval histogram of GVI value within the study zone.

### 3.2. Overall Spatial Distribution Characteristics of GVI for Some of the Roads within the Second Ring in Changsha

The GVI spatial distribution result of partial roads within the Second Ring in Changsha is shown in Figure 9. The study area was divided into two parts, Hedong and Hexi regions, with the Xiang River as the boundary. Overall, the GVI spatial distribution shows the characteristics of increasing from the Second Ring center to the boundary, and the Hexi region has a higher GVI than the Hedong region. There are far more street points with a low-GVI than that with high-GVI in the study area, and street points with GVI higher than 35% show a scattered distribution (Figure 10). In the Hedong region, the number of street

points with a GVI lower than 5% decreases outward from the intersection of Wuyi Road and Furong Road(M), showing a trend of a low center and high periphery, while the Hexi region shows a scattered distribution characteristic (Figure 11).

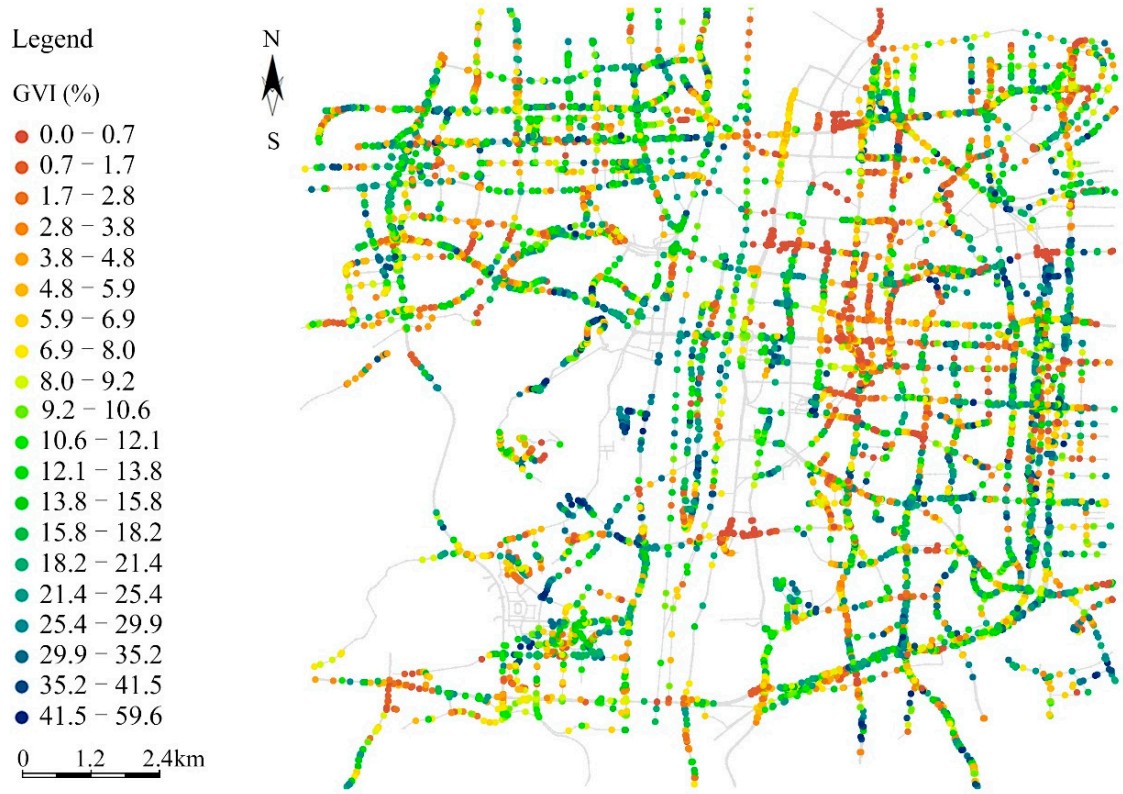

**Figure 9.** Spatial distribution of roads GVI.

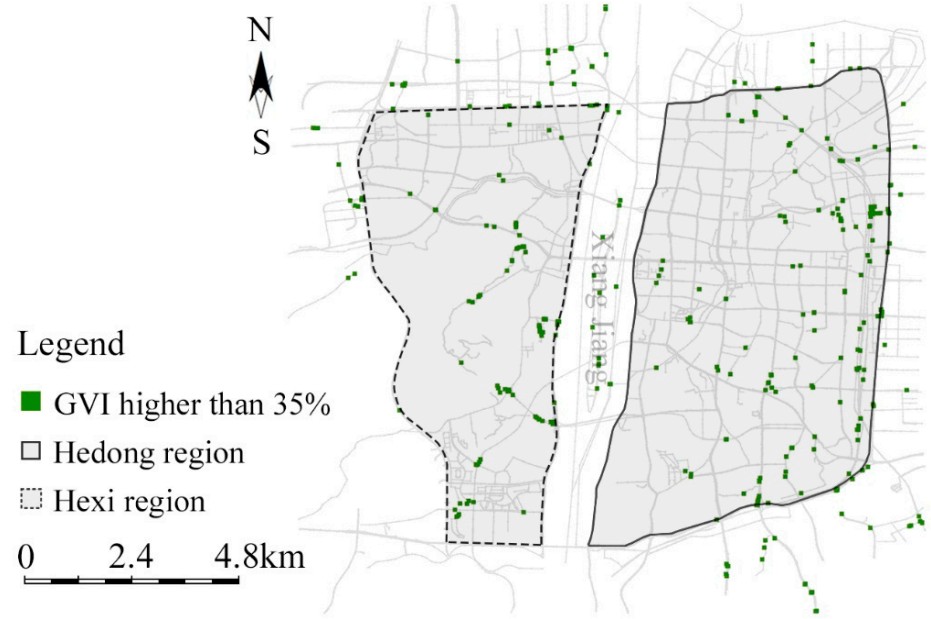

**Figure 10.** Distribution of street points with higher than 35% GVI.

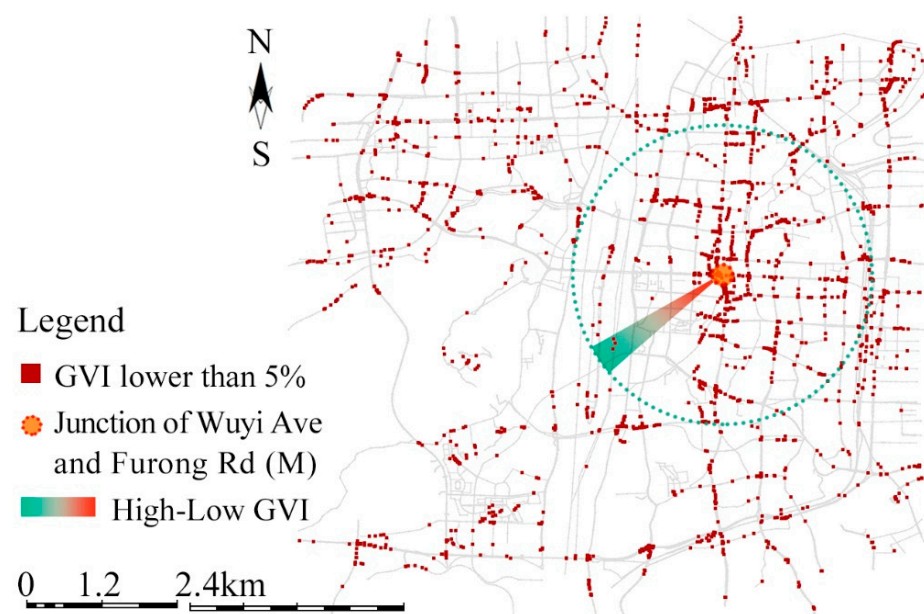

**Figure 11.** Distribution of street points with lower than 5% GVI.

*3.3. Characteristics and Comparison on GVI of Partial Roads in Hexi Region and Hedong Region within the Second Ring in Changsha*

After statistics and analysis, the interval percentage of average GVI in the Hexi and Hedong regions within the Second Ring in Changsha are shown in Figure 12. The length of the light green square bar represents the proportion of the corresponding GVI interval in the Hexi region. In contrast, the length of the dark green one represents the proportion of the corresponding GVI interval in the Hedong region. Most areas in both regions have a range of 0–45% of GVI intervals. The highest GVI interval proportion in the Hexi region is 5–10%, compared to 0–5% in the Hedong region. The average GVI of partial roads in the Hexi region is 13.59%, of which 19.5% are streets with very poor green perception, 46.9% are streets with poor perception, 19.6% are streets with general perception, 9.2% are streets with strong perception, and 4.8% are streets with very strong perception. On the other hand, the average GVI of partial roads in the Hedong region is 11.93%, of which 32.1% are streets with very poor green perception, 35.2% are streets with poor perception, 18.9% are streets with general perception, 10.3% are streets with strong perception, and 3.5% are streets with very strong perception.

In comparison, except for the 0% GVI point and 0–5% GVI intervals, each GVI interval in the Hexi region accounts for more than that in the Hedong region, which shows the level of greenery in the Hexi region is better than that in Hedong region. The reason is probably that the Hexi region has more tourist attractions and green parks, such as Yuelu Mountain, West Lake Park, and Wang Yue Park, which focus on green landscape creation. Unlike the Hexi region, the Hedong region has more urban core business circles, such as Wuyi, Dongtang, and Wujialing. These areas have dense commercial buildings, narrow sidewalks, and streets greening mainly with trees lining the street, which are not conducive to urban greenery development. Yet from a general perspective, the average street GVI in both the Hexi and Hedong regions within the Second Ring is still at a low level, with a large proportion of very poor green perception and poor green perception, accounting for 66.4% and 67.3%, respectively, as well as reflecting that the overall greenery level of streets in both the Hexi region and Hedong region within the Second Ring needs to be improved.

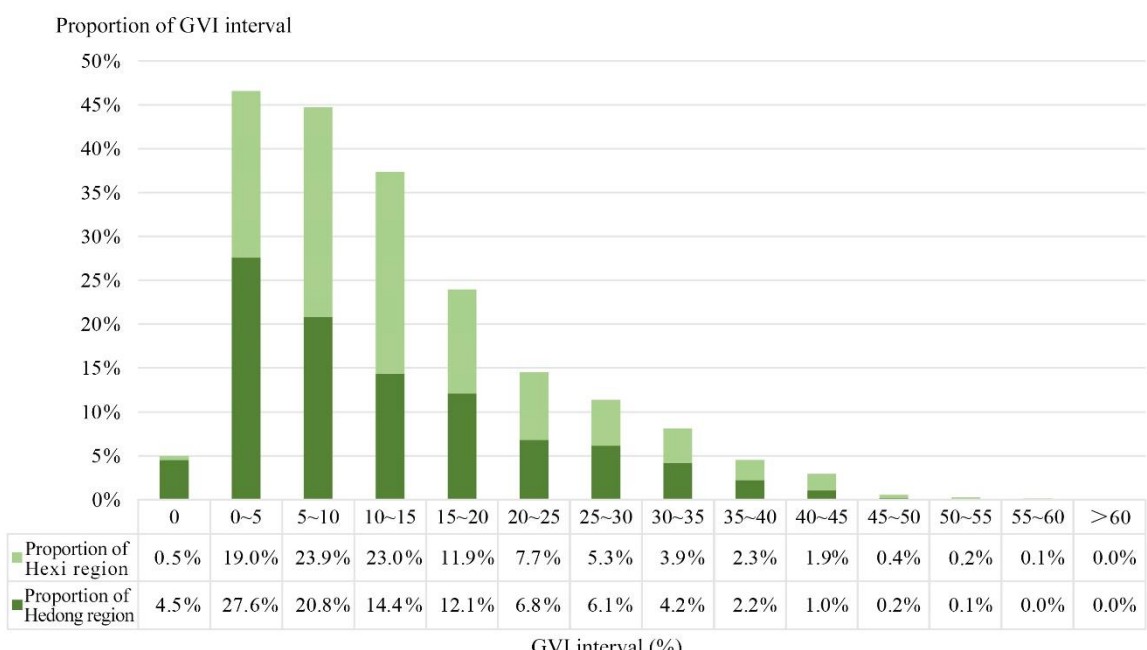

Proportion of GVI interval

| | 0 | 0~5 | 5~10 | 10~15 | 15~20 | 20~25 | 25~30 | 30~35 | 35~40 | 40~45 | 45~50 | 50~55 | 55~60 | >60 |
|---|---|---|---|---|---|---|---|---|---|---|---|---|---|---|
| Proportion of Hexi region | 0.5% | 19.0% | 23.9% | 23.0% | 11.9% | 7.7% | 5.3% | 3.9% | 2.3% | 1.9% | 0.4% | 0.2% | 0.1% | 0.0% |
| Proportion of Hedong region | 4.5% | 27.6% | 20.8% | 14.4% | 12.1% | 6.8% | 6.1% | 4.2% | 2.2% | 1.0% | 0.2% | 0.1% | 0.0% | 0.0% |

GVI interval (%)

**Figure 12.** The percentage of GVI interval in different regions within the Second Ring in Changsha.

*3.4. Comparative Analysis of the GVI Characteristics in Each Administrative District and Each Street within the Second Ring in Changsha*

The average GVI of partial roads in each administrative district within the Second Ring in Changsha (Figure 13) showed that the order from high to low was 14.2% in Yuhua District, 14.0% in Yuelu District, 12.4% in Furong District, 10.3% in Tianxin District, and 9.7% in Kaifu District.

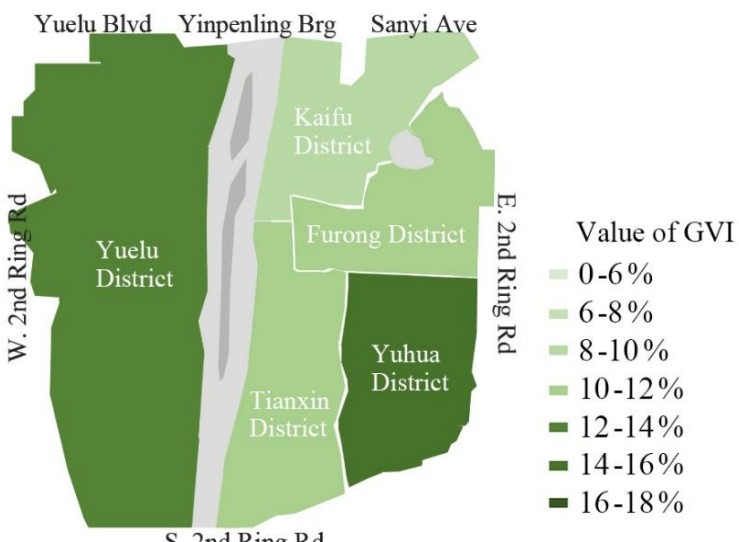

**Figure 13.** Comparison of the average level of GVI in each administrative district within the study area.

There are 28 sub-district offices in the five administrative districts within the study area, and their average GVIs are shown in Figure 14. Among all the streets, there is one street with a GVI of less than 5% (very poor green perception), which is Xiangya Road Street in Kaifu District; five streets with a GVI of 15% to 25% (general green perception), which are Shazitang Street and Houjiatang Street in Yuhua District, Pozi Street and Xianghu

Street in Furong District, and Orange Island Street in Yuelu District; most of the streets have a GVI of 5–15% (poor green perception), totaling 22 streets.

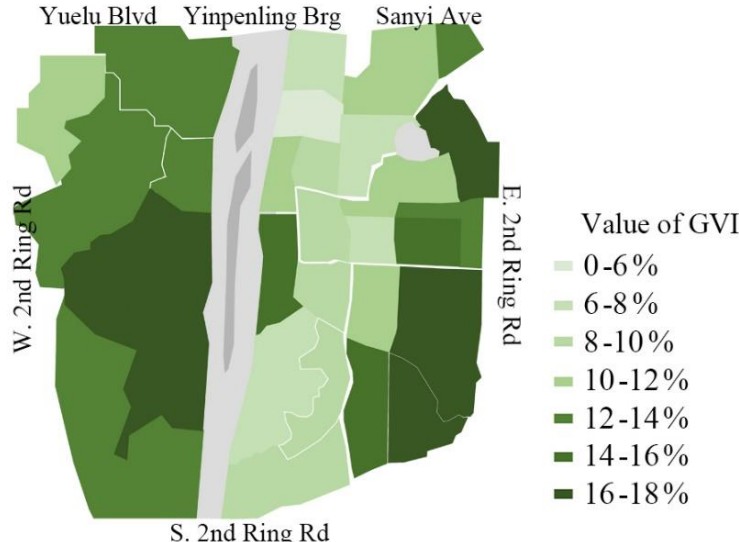

**Figure 14.** Comparison of the average green view level on each street in the study region.

In terms of the GVI characteristics in each administrative district and street, it is generally in line with "Commerce East, Culture West", the urban function layout in Changsha. Furong District (Figure 15) and Tianxin District (Figure 16) in the Hedong region are the central old urban areas, and their western parts are close to the Ring center, in which many old commercial areas exist. As a result, the commercial buildings gathered, and the greenery construction is insufficient, while their eastern parts are near Second Ring Road, and include more green space and waterscape, such as Martyr Memorial Park, Jiudaowan Community Park, Wangfu Park, and Xiaoyuan Park. Therefore, the three streets on the eastern parts of Furong District, Xiaohu Street, Wulipai Street, and Chaoyang Street, have a higher level of GVI than that of the western street, and are also higher than the average level within the Second Ring.

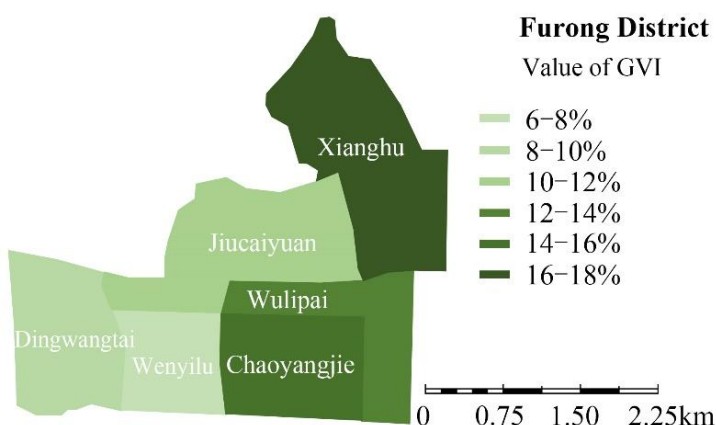

**Figure 15.** Average GVI of each street in Furong District.

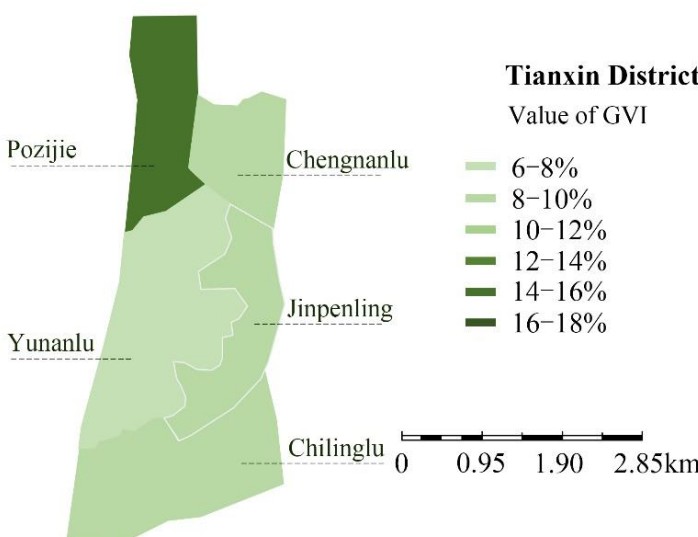

**Figure 16.** Average GVI of each street in Tianxin District.

Yuhua District (Figure 17) is a new urban area built by expanding outward from the center of Hedong region, with a better level of greening than the old urban area. Yuelu District (Figure 18) is a typical representative of the "Mountain and Water City" with its ecological landscapes such as Yuelu Mountain, Houhu Lake, and Meixi Lake. Therefore, the overall GVI of Yuhua District and Yuelu District is higher than the average level within the Second Ring, and the streets with higher GVI within the study area are also concentrated in these two districts.

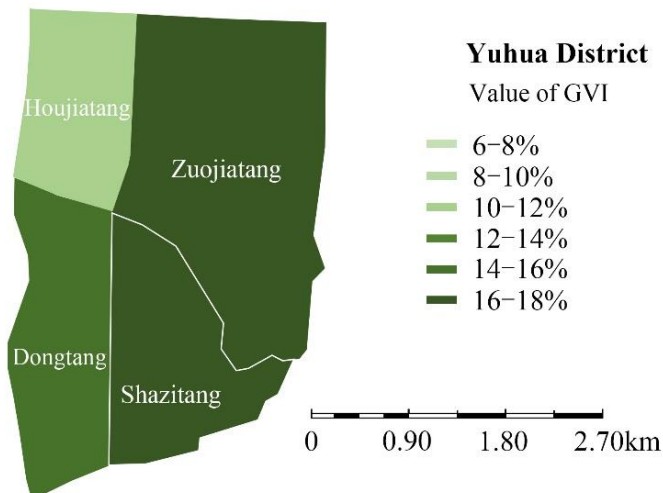

**Figure 17.** Average GVI of each street in Yuhua District.

Kaifu District has a pattern of "New North and Old South" (Figure 19). Within the study area, Kaifu District is a part of the old south urban area, which has the lowest level of GVI among the administrative districts. Only Sifangping Street (within the study area), which is close to the periphery of the Second Ring, is higher than the average level. The GVI spatial distribution in each street also reflected that the GVI was worse as it goes inside the Second Ring and better as it goes outside the Second Ring.

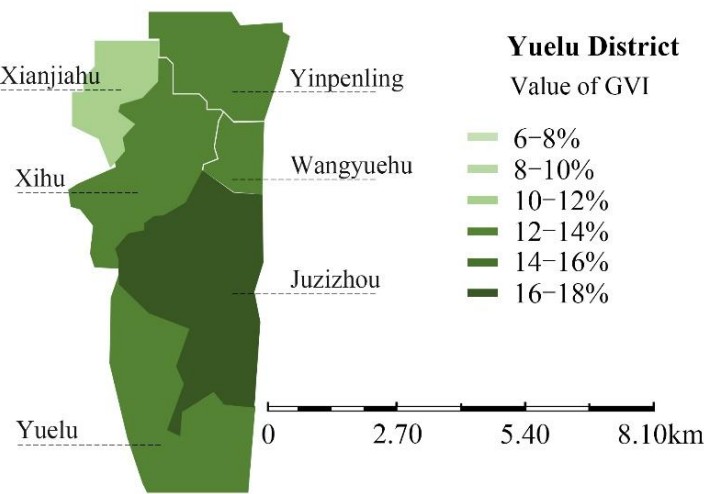

**Figure 18.** Average GVI of each street in Yuelu District.

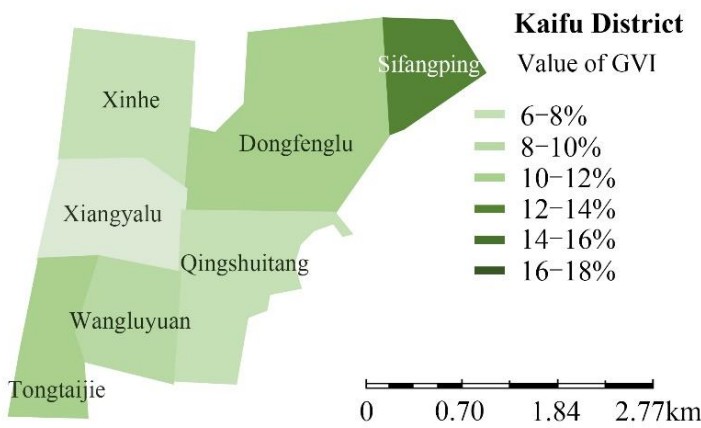

**Figure 19.** Average GVI of each street in Kaifu District.

## 4. Improvement Suggestions

The overall data shows that Changsha City has a low-GVI level within the Second Ring. In order to make the GVI of urban roads in Changsha improve effectively in the future, appropriate suggestions need to be proposed from the overall planning perspective. In general, for areas with low-GVI, focusing on planting design, paying attention to the different height collocation of plants, and looking for a reasonable combination of trees, shrubs, and herbs, are three effective ways to create an order conversion in landscape space. In addition, it's necessary to choose plant types that change with the seasons to enrich the landscape space layers and diversity.

For wide roads in the city center area, the road center line or boundary of different road levels are arranged with vegetation to ensure that people can see the greenery from both sides of the sight line when they move on the sidewalk, to enhance the overall GVI of the roads. The sidewalk is arranged with taller trees to isolate noise, and at the same time, improve the road greening level; smaller trees could be arranged between the car and bike lanes on both sides.

Due to historical planning and other reasons, the old streets in the urban center have no extra land for improvement, and the roads with narrow sidewalks can only or cannot plant trees, with much less space to plant shrubs. Therefore, vertical greening should be arranged in combination with surrounding buildings, such as planting climbing plants, designing plant landscape walls, and placing or hanging potted plants. In addition, developing facilities' greening can improve the GVI of streets by compounding the utilization of

space; the old communities should achieve organic renewal to enhance the greening of the surrounding roads.

The current method aggerated tree, grass, and vegetation segmentation results to calculate GVI. In this case, the greenery information from different orientations and sources had the same importance as long as they were shown in the street view images. The greenery information from different orientations and sources may have different importance for pedestrians, and some may be unhelpful for walking pedestrians. In the future, it is necessary to enhance the software to determine the importance of different greenery sources for GVI calculation.

The individual GVI value calculated using fish-eye-type 360-degree images may differ slightly from human vision. On the other hand, the image overlap problem may occur when using normal images from different orientations. In this study, the fish-eye-type 360-degree images were used to capture the greenery information from all orientations and avoid overlapping image problems. However, comparing GVI values from different regions is still meaningful because all semantic identification was made with the same image type.

## 5. Conclusions

In this paper, the overall GVI analysis of urban roads in Changsha was conducted based on a semantic segmentation neural network with SVI data in Baidu Map. The results show that: (1) The average GVI of the main roads within the Second Ring in Changsha is 12.56%, which is a low level of greening and still has much progress to be made. (2) In terms of spatial distribution, the GVI of the Hedong region is 11.26%, and that of the Hexi region is 13.61%, with the overall characteristics of "High West and Low East". The reason for higher GVI in the Hexi region may be that the greening construction in the Hexi region, such as tourist attractions, water parks, and community parks, is of higher quality. The old commercial center and the railway station are often the main factors leading to the low GVI of the roads in the Hedong region. (3) In the administrative districts, the highest GVI within the Second Ring is Yuhua District with 14.15%, the lowest is Kaifu District with 8.75%, and the districts in order of descending GVI are: Yuhua District, Yuelu District, Furong District, Tianxin District, and Kaifu District. The GVI of the new urban area is better than that of the old urban area, which is affected by the distribution of buildings, with higher building density and lower GVI.

Due to the limited computational performance of the equipment used, the number of sampling points collected is relatively small for an urban-scale study, which has an impact on the accuracy of the conclusions. There is also the use of fish-eye pictures that can cause deviations from the real situation and may have an impact on the accuracy of the data itself to some extent. Moreover, the network street view images are street view data collected by collection vehicles in the middle of the roads, which deviates from the real feelings of pedestrians on the road and has an impact on the objectivity of the data. In addition, the Baidu Street View data used for the study was taken and uploaded in the summer of 2019; thus, the effect of seasonal changes on the GVI of the streets was ignored. In order to improve the accuracy of the conclusions, the data needs to be collected as comprehensively as possible in future experiments.

**Author Contributions:** Conceptualization, Y.C.; Methodology, Y.C.; Software, Z.D.; Formal analysis, X.F., Z.X., X.K., K.P. and Z.G.; Investigation, Z.D., X.F., Z.X., X.K., K.P. and Z.G.; Resources, X.K.; Data curation, Z.D., X.F., Z.X., X.K., K.P. and Z.G.; Writing—original draft, Q.Z., X.F. and Z.X.; Visualization, K.P.; Supervision, Y.C.; Project administration, Z.D.; Funding acquisition, Y.C. All authors have read and agreed to the published version of the manuscript.

**Funding:** This research was funded by Hunan University, China, through Course Development Program of "Artificial Intelligence in Built Environment".

**Institutional Review Board Statement:** Not applicable.

**Informed Consent Statement:** Not applicable.

**Data Availability Statement:** Not applicable.

**Conflicts of Interest:** The authors declare no conflict of interest.

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
