# Peer review of "Research on Green View Index of Urban Roads Based on Street View Image Recognition: A Case Study of Changsha Downtown Areas"

_sustainability, doi:10.3390/su142316063_

Round 1

Reviewer 1 Report

Dear Authors,

This paper is well written; however, I have a few comments:

1.     The aim/contributions/objectives or research questions should be stated clearly.

2.     All figures should be enlarged and made clear.

3.     The authors should explain more in figure 12.

Author Response

We authors sincerely thank all reviewers for your valuable comments. The feedback has helped us improve the manuscript's deficiencies and increase the article's rigor and readability. We have completed the revision using track changes and have responded to your comments point by point. Please see the attachment for details. 

Reviewer 2 Report

I thank the editors for the opportunity to review this manuscript

The authors have used a previously described method to determine the GVI of urban roads in Changsha. The authors’ premise of the GVI being a significant contributor to the livability of urban built up areas. However, the authors further mention that the data involves greenery involving road dividers which is a confounding factor. Would it be possible for the authors to train the analytical software to ignore the greenery that is perceived to be unhelpful in this aspect by pedestrians? This would be possible by identifying segments of roads with these dividers and a useful piece of information. 

With the use of a fish eye type camera to capture as much of a view, images are inevitably distorted. Is there a correction method involved in ensuring that pixels are accurately represented size-wise as this may influence the percentage of the representative image. 

Would suggest the authors relook and restructure sentences (line 30-42 for instance), and reword content presented to aid in readability. Further odd expressions include”vegetation ‘in’ the street” which should be ‘lining’ instead. Subtle expressions such as this do affect the impression of the viewership.

A few more examples have been highlighted below

Sentence 89-94 does not make sense and needs to be reworded.

Line 100 - is exactly the right word?

Line 182 - “we take the batch and precise method”. What does this mean?

Author Response

We authors sincerely thank all reviewers for your valuable comments. The feedback has helped us to improve the deficiencies in the manuscript and increase the rigor and readability of the article. We have completed the revision using track changes and have responded to your comments point by point. Please see the attachment for details.

Round 2

Reviewer 2 Report

I thank the authors for addressing the comments provided. The quality of writing has been improved, and is more holistic given incorporating of discussions surrounding the limitations of this approach.